# LAPLACIAN DENOISING AUTOENCODER

## ABSTRACT

While deep neural networks have been shown to perform remarkably well in many machine learning tasks, labeling a large amount of supervised data is usually very costly to scale. Therefore, learning robust representations with unlabeled data is critical in relieving human effort and vital for many downstream applications. Recent advances in unsupervised and self-supervised learning approaches for visual data have benefited greatly from domain knowledge. Here we are interested in a more generic unsupervised learning framework that can be easily generalized to other domains. In this paper, we propose to learn data representations with a novel type of denoising autoencoder, where the noisy input data is generated by corrupting clean data in the gradient domain. This can be naturally generalized to span multiple scales with a Laplacian pyramid representation of the input data. In this way, the agent learns more robust representations that exploits the underlying data structures across multiple scales. Experiments on several visual benchmarks demonstrate that better representations can be learned with the proposed approach, compared to its counterpart with single-scale corruption. Furthermore, we also demonstrate that the learned representations perform well when transferring to other vision tasks.

## 1 INTRODUCTION

In recent years, deep learning has made significant improvements on machine learning tasks. However, the success of deep-based approaches relies greatly on using a large amount of human labeled data for supervision, which is usually very costly and infeasible to scale on new data. Actually, humans are exceptional experts at learning abstract knowledge from unsupervised data, *i.e.*, without knowing the specific labels of the data. Thus, how to imitate such a human cognitive ability and effectively learn robust representations from massive sums of unlabeled data in an unsupervised manner have attracted the interests of machine learning researchers.

Representation learning is a popular framework for unsupervised learning that aims to learn transferable representations from unlabeled data (Bengio et al., 2013). Although great progress has been achieved for visual data by some recent advances (Zhang et al., 2016; 2017; Pathak et al., 2016; Noroozi & Favaro, 2016; Doersch et al., 2015; Noroozi et al., 2017; Pathak et al., 2017; Gidaris et al., 2018), the approaches are mostly designed to boost the performance of high-level recognition tasks like classification and detection. We argue that good representations should benefit multiple kinds of tasks, including both high-level recognition tasks and low-level pixel-wise prediction tasks. We, in this paper, present a novel unsupervised representation learning approach that is applicable to more generic type of data and tasks. The only assumption about the input data form is that the learned representations should incorporate the underlying data structures along some certain dimensions. For example, one would expect the representations for visual data to incorporate underlying image structures along the spatial dimension, while the representations for speech data might need to be exploited along the temporal dimension.

Specifically, we propose to decouple the representations into different semantic levels in the Laplacian domain. A novel type of denoising autoencoder (DAE) (Vincent et al., 2010) is proposed to distill both high- and low-level representations accordingly. Different from the conventional DAE, where the noisy input is generated from the clean data by adding noises in the original space, we propose to generate noisy input by corrupting the clean data in the gradient domain. By perturbing the clean data in such a manner, the corruptions are diffused into larger scales and made more difficult to remove (Fig. 1 (a)). More importantly, the gradient domain corruption can be naturally

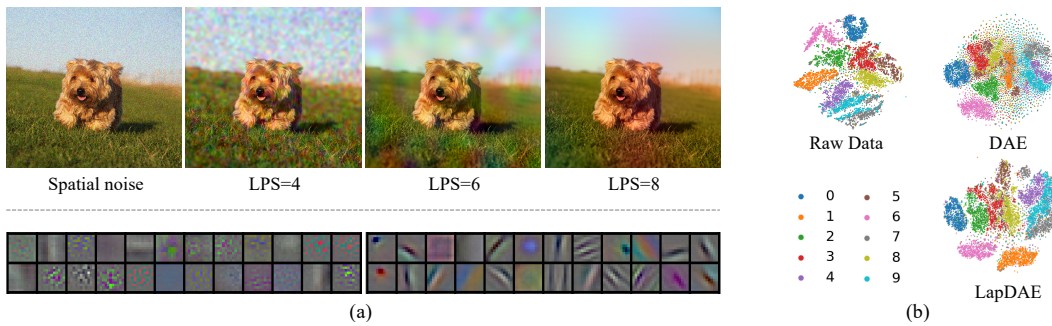

Figure 1: (a) Top: illustration of our Laplacian pyramid based corruption construction strategy compared to traditional spatial corruption, where "LPS" indicates the Laplacian pyramid scale; Bottom: learned kernels when corruption is added in spatial domain (left) and gradient domain (right). (b) Visualization of the discriminative capability on the MNIST test dataset, where samples are projected to the 2D domain by using the t-SNE technique (Maaten & Hinton, 2008).

extended to span multiple scales with a Laplacian pyramid representation of the data (Burt & Adelson, 1983). To this end, the DAE is enforced to learn more robust representations that can exploit the underlying data structures across multiple scales. In addition, the proposed learning approach can easily be incorporated into other representation learning frameworks, and boosts their performance accordingly.

Our motivation is inspired by the human knowledge learning by visual perception. Instead of trying to remember every single detail, human vision focuses more on the general concept of the object/scene, which favors a combined perception of both local and non-local information (Bubić et al., 2010). An example of the proposed gradient-domain corruption is illustrated in Fig. 1 (a). It can be observed that compared to directly adding noise spatially, editing on different scales of the Laplacian pyramid leads to non-local random corruptions. We also show the learned kernels by the corruption in the spatial domain and that in the gradient domain at the bottom of Fig. 1 (a). It can be observed that more edge-sensitive and color-sensitive kernels and non-local responses are learned by the gradient domain corruption (right) in comparison to spatial corruption (left) which preferring local responses. We argue that in order for an agent to be able to recover the corruptions from different scales non-locally, it requires an understanding of the context in the scene.

In Fig. 1 (b), we illustrate the discriminative capability of our model on the MNIST (LeCun et al., 1998) testing set. The visualization is achieved by projecting the high-dimensional data or feature to a two-dimension space, using the t-SNE (Maaten & Hinton, 2008) technique. Compared to the raw data distribution, the embedding space of the conventional denoising autoencoder shows a better clustering ability while with some background noise. When compared to the embedding of the proposed Laplacian denoising autoencoder (LapDAE), we can observe that different categories (digits) are well discriminated from each other and with much less noise. For example, the digit *5* and *3* are better discriminated compared to those from the raw data and from DAE.

We demonstrate the effectiveness of the proposed unsupervised learning approach in two folds: 1) by evaluating the clustering and discriminative capability on classic benchmarks; 2) by training on large-scale data and transferring the learned visual features to a variety of vision tasks including multi-label classification, object detection, and semantic segmentation. The main contributions of our work are summarized as follows:

- We propose a new unsupervised representation learning framework , by enforcing the model to learn more context and discriminative information in the Laplacian domain.

- The proposed framework is trained only based on the raw data itself and neither the data domain assumptions nor pseudo labels are necessary.

- Our framework is superior to the conventional DAE and achieves state-of-the-art performance on several benchmarks.

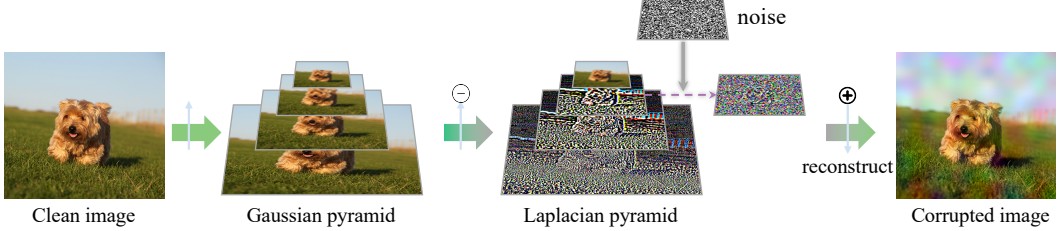

Figure 2: The illustration of the corruption with a Laplacian pyramid. A Gaussian pyramid is first constructed from the clean image, from which a Laplacian pyramid is built. After randomly selecting a level (slice) from the Laplacian pyramid and adding random corruptions (*e.g.*, noise), the final corrupted image is obtained by a reconstruction from the modified Laplacian pyramid.

## 2 RELATED WORKS

**Autoencoders:** The conventional autoencoder (AE) (Hinton & Salakhutdinov, 2006) is based on the idea of learning a mapping from high-dimension to low-dimension so that the encoded representation can be used to reconstruct the original raw input. Bengio et al. (2007) propose to learn the abstract representation by stacking single-layer autoencoders. Poultney et al. (2007) impose a sparsity prior for the latent encoded space. Furthermore, the denoising autoencoder (DAE) (Vincent et al., 2010) is proposed to achieve abstraction that is robust to noise and is proven to be able to learn better representations. The variational autoencoder (VAE) (Kingma & Welling, 2013) aims to learn a parametric latent variable model by encouraging the latent space to satisfy a distribution. We refer to (Bengio et al., 2013) for a broader view of autoencoder-based approaches.

**Representation learning:** As a fundamental problem, representation learning has been studied for years. A comprehensive review could be referred to (Bengio et al., 2013). Early classical algorithms mainly focus on reconstructing the raw data by learning compressed features (Hinton & Salakhutdinov, 2006; Vincent et al., 2010). Some other methods use probabilistic models like Boltzmann machines (Hinton & Sejnowski, 1986; Salakhutdinov & Larochelle, 2010) and GANs (Goodfellow et al., 2014; Donahue et al., 2016). Recently, some approaches address the problem by defining some pretext tasks (termed "self-supervised learning") which have shown promising performance, including predicting relative spatial location/ordering of image patches (Doersch et al., 2015; Noroozi & Favaro, 2016; Noroozi et al., 2017), motion in video (Misra et al., 2016; Pathak et al., 2017; Wang & Gupta, 2015), colorization (Zhang et al., 2016; 2017), predicting rotations (Gidaris et al., 2018) and transformations (Zhang et al., 2019), to name a few. Such pretext-task-based methods can be categorized into a different group compared to the AE/DAE-based ones. Compared to these representation learning methods, the proposed approach does not make any assumptions to the data thus is more generic.

## 3 LAPLACIAN DENOISING AUTOENCODER

### 3.1 BACKGROUND

The original Laplacian pyramid was proposed for image editing (Burt & Adelson, 1983) and can easily be generalized to other types of data where a low-pass filter is applicable. Given input data $x$, its Gaussian pyramid is composed of a set of progressively lower resolution versions of the data, denoted as $\{x_l^G\}$ where $l$ is a pyramid level. In the pyramid, the bottom level is the data itself, $x_0^G = x$, and $x_{l+1}^G = \text{downsample}(x_l^G)$. The Laplacian pyramid $\{x_l^L\}$ is constructed by subtracting the neighboring levels in the Gaussian pyramid, $x_l^L = x_l^G - \text{upsample}(x_{l+1}^G)$. Note that the top level of the Laplacian pyramid is the residual and the same as that in the Gaussian pyramid, $x_N^L = x_N^G$, where $N$ is the top level of the pyramid. The construction process is shown in Fig. 2. Given a Laplacian pyramid, the original data can be reconstructed by recursively applying $x_l^G = x_l^L + \text{upsample}(x_{l+1}^G)$ until $x_0^G$ is reached. Gradient domain editing on $x$ can be achieved by editing its Laplacian pyramid and then reconstruct the resulting $\tilde{x}$ from the modified Laplacian pyramid.

## 3.2 LapDAE Methodology

Following the denoising autoencoder framework (Vincent et al., 2010), we attempt to distill the essential representations by training a convolutional network (ConvNet) to restore the clean data $x \in \mathcal{X}$ from the corrupted data $\tilde{x} \in \tilde{\mathcal{X}}$. In contrast to a standard DAE, we generate the corrupted data $\tilde{x}$ from $x$ with the aid of a Laplacian pyramid. Specifically, we construct a Laplacian pyramid from the clean data $x$ and randomly corrupt a level of the pyramid, such that $\tilde{x}$ is reconstructed from the corrupted pyramid. Fig. 2 illustrates the process of the corruption with an example of image data. Since the corruption applied to higher levels of the pyramid affects larger spatial scales of the image (see Fig. 1), the randomly corrupted levels will enforce the network to learn features that can represent underlying structures across multiple scales. As known in the literature (Zeiler & Fergus, 2014), the ConvNet is inherently in favor of both local and non-local features at different levels of layers. Hence, with only local disturbances, it is difficult to capture the non-local semantic concepts. This has also been verified to some extent in the self-supervised learning methods that attempt to leverage patch-based context information (Doersch et al., 2015; Noroozi & Favaro, 2016; Noroozi et al., 2017), and similarly to the non-local scheme (Wang et al., 2018) on the network design side. By adding corruptions across multiple scales, the objective is to capture both local and non-local information during the representation learning phase. Additionally, in order to incorporate diverse types of corruptions and to force the network to "learn harder", it is also possible to apply multiple types of corruptions to the pyramid during learning.

Denote the corrupted data as $\tilde{x} = Lap(x; \widetilde{x_l})$ (note that we use $\widetilde{x_l}$ instead of $\widetilde{x_l^L}$ for conciseness to denote a corrupted level $l$ in the Laplacian pyramid). The corrupted input space is achieved as $\mathcal{L} = \{Lap(x; \widetilde{x_l})_c, c \in \mathcal{C}\}$, where $\mathcal{C}$ is the corruption type set. Then the corrupted space is mapped to a hidden space $\mathcal{Y} = F_\theta(\mathcal{L})$ through an encoder with parameter $\theta$. Differing from a DAE that uses sparse code as a hidden representation, each sample in our hidden space has its own resolution, from which we recover the reconstruction space $\mathcal{Z} = G_{\theta'}(\mathcal{Y})$ with a decoder with parameter $\theta'$. During the optimization in mini-batch, each time a training sample $x$ is presented, one or multiple versions of corruptions are constructed according to $\mathcal{C}$. Therefore each time the optimization is based on a sub-mini-batch, for which a sub-batch reconstruction objective is defined as:

$$L_{\text{rec}} = \sum_{c \in \mathcal{C}} \mathbb{E}_x \parallel x - z_c \parallel_2^2, \qquad (1)$$

---

**Algorithm 1:** LapDAE Optimization

Initialize corruption set $\mathcal{C}$ and parameters $\theta, \theta'$
**while** *not converged* **do**
    **for** $x \in \mathcal{X}$ **do**
        Compose the Gaussian pyramid $\{x_l^G\}$
        Construct the Laplacian pyramid
        $\{x_l^L\}$ from $\{x_l^G\}$
        **for** $c \in \mathcal{C}$ **do**
            Randomly select a pyramid level $l$
            $\widetilde{x_l} \leftarrow$ apply corruption $c$ on level $l$
        **end**
        Reconstruct the corrupted data
        $\tilde{x} = Lap(x; \widetilde{x_l})$ in image domain
        $y = F_\theta(\tilde{x})$
        $z = G_{\theta'}(y)$
    **end**
    $\min \sum\limits_{x \in \mathcal{X}} \sum\limits_{c \in \mathcal{C}} \parallel x - z_c \parallel_2^2$
**end**
Return $\theta, \theta'$ for the LapDAE

---

where $z_c \in \mathcal{Z}$ is the reconstructed data by the network. The learning process of the proposed LapDAE is summarized in Algorithm 1.

The proposed Laplacian denoising autoencoder performs data reconstruction in the Laplacian pyramid space across multiple scales. Despite simple and making no assumptions about the data and requiring no specially designed domain-specific loss functions, the proposed framework is able to learn representations competitively with existing unsupervised (as well as some self-supervised) approaches. This will be shown in the following evaluation section. Similar to some recent work which has explored withholding parts of the data (*e.g.*, AE to remove noise; inpainting and context-encoder to drop data in spatial domain; colorization to drop data along channel direction), our LapDAE model can be considered as removing context-aware *noise* along the scale direction in Laplacian domain.

Being a purely unsupervised model, this is a generic framework that can be applied to other domains in addition to visual data. The proposed framework opens a new potential direction for representation learning in another transferred domain (*e.g.*, gradient domain), which we believe to be beneficial

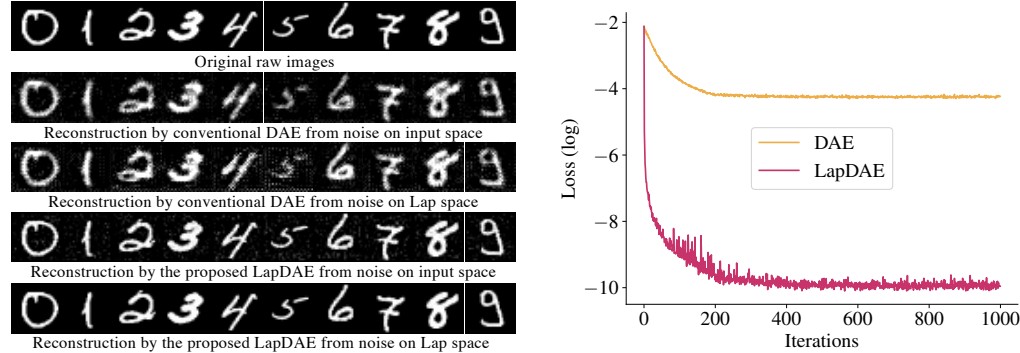

Figure 3: Left: An illustration of the reconstruction performance on the MNIST dataset. The original raw input images are randomly selected from the test set and are shown in the first row, while the second and last rows show the reconstructed results from conventional DAE and our LapDAE, respectively. DAE applied on Lap noise space and LapDAE on spatial noise space are also shown for reference in the third and fourth rows. Right: Illustration on model convergence. The horizontal axis shows the training iterations while vertical axis the training loss (in log scale).

to the community, in which current work focuses mainly on knowledge mining in the original (spatial) domain.

### 3.3 LapDAE Architecture

In this section, we describe the detailed architecture of the proposed LapDAE. Specifically, we utilize a convolutional neural network (CNN) to implement the LapDAE and showcase on visual data. The corruption in the Laplacian domain is modeled as a Laplacian layer. Specifically, we randomly add Gaussian noise (with $\sigma = 25$) to a randomly chosen level $l$ in the Laplacian pyramid. The encoder consists of several simple convolutional (*conv*) layers, while the decoder is of a mirrored structure to the encoder and consists of *up-conv* (also termed *deconv* in some literature) layers. The model is trained with supervision from the reconstruction objective defined in Equation (1).

## 4 Experiments

### 4.1 Experimental Setup

The setup of the proposed framework is described in Sec. 3.3. For the basic LapDAE architecture, four $3 \times 3$ *conv* layers are used to construct the encoder, while the decoder consists of three similar *up-conv* layers. For the experiments performed on large-scale datasets (Sec. 4.4, 4.5), we use the AlexNet (Krizhevsky et al., 2012) structure for the encoder and similarly the decoder consists of three *up-conv* layers. For simplicity in this study, only one corruption type, random noise, is set in $\mathcal{C}$. The Laplacian pyramid is constructed with eight levels. The whole model is trained end-to-end by the Adam optimizer (Kingma & Ba, 2014), with the learning rate set to $10^{-4}$. The learning rate decreases at a factor of $10^{-1}$ for every 20 epochs. A batch size of 128 is used throughout the experiments.

### 4.2 Evaluation on MNIST

The MNIST [1] dataset consists of 70,000 images of handwritten digits with size of $28 \times 28$, in which 60,000 are used for training and the rest 10,000 for testing. Randomly selected example images from the MNIST are shown in Fig. 3 (left). In this experiment, the input images are fed into the model at fixed size of $28 \times 28$ with only horizontal flipping as data augmentation during training. As the objective for our model is set as the reconstruction error (as in Equation 1), we first illustrate the qualitative performance on the image reconstruction, shown in Fig. 3 (left). As we can see from the

---

[1] http://yann.lecun.com/exdb/mnist/

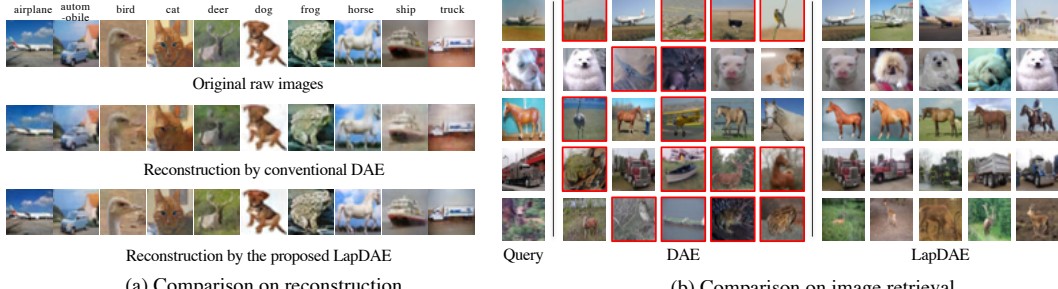

(a) Comparison on reconstruction  (b) Comparison on image retrieval

Figure 4: Evaluation using the CIFAR-10 dataset. (a) The reconstruction result comparison, in which examples from each category are randomly selected for visualization. (b) The image retrieval results by nearest neighbor. Given the query on the left, the top-5 (from left to right) retrieved results of DAE (middle) and LapDAE (right) are presented, in which red the ones indicate wrong category.

reconstruction results, the conventional DAE generally reconstructs the digits but they are unclear and include some noise. In contrast, the reconstruction of our LapDAE model is evidently much clearer and includes more details, *e.g.*, the numbers *0* and *4*. To better understand the reconstruction capability, we apply the conventional DAE to images corrupted with Laplacian noise, with comparison to applying our LapDAE to images where the noise is added on the input space. The results again suggest that the proposed LapDAE performs better on reconstructing context information, *i.e.*, digits here. We also compare the convergence property of the proposed LapDAE and conventional DAE, as shown in Fig. 3 (right). It can be observed that with the proposed LapDAE, the model converges faster and results in reaching a much more optimum level.

We perform an experiment on image clustering for both DAE and the proposed LapDAE models. The result is shown in Fig. 1 (b). From the results, we can see that the proposed LapDAE has a far better discriminative capability compared to the conventional DAE.

### 4.3 EVALUATION ON CIFAR

In comparison to the MNIST dataset, the CIFAR-10 (Krizhevsky, 2009) dataset is composed of RGB natural images with a size of $32 \times 32$, covering 10 different categories of natural objects. The training set consists of 50,000 images while the testing set is 10,000. Each category includes 6,000 images. In this experiment, we first visualize the reconstructed images and compare to those reconstructed by the conventional DAE, see Fig. 4 (a). From the results we can see that by using the proposed LapDAE, the representative context is well reconstructed, *e.g.*, the face of the *cat* and the *deer* in the forest. We also explore the quality of the learned representation, by performing an image retrieval task. The retrieval is based on the similarity of the embedding space, by using the nearest neighbor scheme. Given an input query image, the feature at the bottleneck (latent space) of the model is extracted for the retrieval in the whole testing set. For this experiment, we compare with results from the conventional DAE. Fig. 4 (b) shows example results. From the results in Fig. 4 (b) we can observe that our LapDAE model learns a much better representation compared to the DAE. The conventional DAE tends to retrieve based on the appearance of the images, while our LapDAE focuses more on the context/semantic information, *e.g.*, the *airplane* in the first row. We attribute this to the multiple scale corruptions in the Laplacian domain. Overall, these results together with the above evaluation on the MNIST dataset can be considered as proof of concept that the proposed LapDAE is capable of capturing both low-level and high-level context information.

### 4.4 EVALUATION ON IMAGENET

In this section, we aim to investigate the representation learning capability of the proposed LapDAE on large-scale data. To achieve this goal, we perform evaluations on the ImageNet (Deng et al., 2009) dataset. Specifically, we use the training set without labels from ImageNet (Deng et al., 2009) to train our LapDAE model. The training set includes 1.2 million images covering 1,000 categories. Each image is first resized to $256 \times 256$ and randomly cropped to 227. Horizontal flipping is also applied for data augmentation.

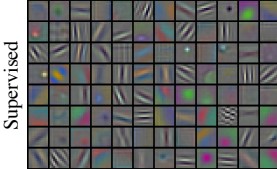 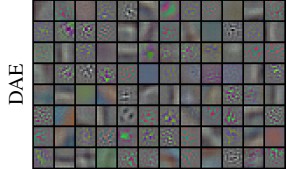 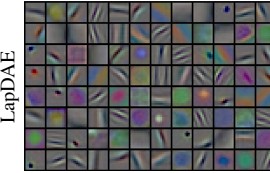

Figure 5: The learned convolutional filters (kernels) from the first layer of AlexNet (*conv1*) trained on the ImageNet dataset. Left: result with fully-supervision from the labeled data; Middle and Right: the filters learned in an unsupervised manner by DAE and our LapDAE respectively.

Table 1: Top-1 accuracy on ImageNet and Places classification with a linear classifier. The results are reported based on the ImageNet / Places validation set and all the listed methods except *ImageNet / Places labels, Domain Adapt* are pre-trained on ImageNet without ground truth labels.

| Method | Conv1 | Conv2 | Conv3 | Conv4 | Conv5 |
|---|---|---|---|---|---|
| ImageNet / Places labels | 19.3 / 22.1 | 36.3 / 35.1 | 44.2 / 40.2 | 48.3 / 43.3 | 50.5 / 44.6 |
| Random Gaussian | 11.6 / 15.7 | 17.1 / 20.3 | 16.9 / 19.8 | 16.3 / 19.1 | 14.1 / 17.5 |
| Random rescaled (Krähenbühl et al., 2015) | 17.5 / 21.4 | 23.0 / 26.2 | 24.5 / 27.1 | 23.2 / 26.1 | 20.6 / 24.0 |
| Context (Doersch et al., 2015) | 16.2 / 19.7 | 23.3 / 26.7 | 30.2 / 31.9 | 31.7 / 32.7 | 29.6 / 30.9 |
| Context Encoder (Pathak et al., 2016) | 14.1 / 18.2 | 20.7 / 23.2 | 21.0 / 23.4 | 19.8 / 21.9 | 15.5 / 18.4 |
| Colorization (Zhang et al., 2016) | 12.5 / 16.0 | 24.5 / 25.7 | 30.4 / 29.6 | 31.5 / 30.3 | 30.3 / 29.7 |
| Jigsaw (Noroozi & Favaro, 2016) | 18.2 / 23.0 | 28.8 / 31.9 | 34.0 / 35.0 | 33.9 / 34.2 | 27.1 / 29.3 |
| BiGAN (Donahue et al., 2016) | 17.7 / 22.0 | 24.5 / 28.7 | 31.0 / 31.8 | 29.9 / 31.3 | 28.0 / 29.7 |
| Split-Brain (Zhang et al., 2017) | 17.7 / 21.3 | 29.3 / 30.7 | 35.4 / 34.0 | 35.2 / 34.1 | 32.8 / 32.5 |
| Counting (Noroozi et al., 2017) | 18.0 / 23.3 | 30.6 / 33.9 | 34.3 / 36.3 | 32.5 / 34.7 | 25.7 / 29.6 |
| RotNet (Gidaris et al., 2018) | 18.8 / 21.5 | 31.7 / 31.0 | 38.7 / 35.1 | 38.2 / 34.6 | 36.5 / 33.7 |
| Domain Adapt (Ren & Lee, 2018) | 16.5 / - | 27.0 / - | 30.5 / - | 30.1 / - | 26.5 / - |
| Instance (Wu et al., 2018) | 16.8 / 18.8 | 26.5 / 24.3 | 31.8 / 31.9 | 34.1 / 34.5 | 35.6 / 33.6 |
| AND (Huang et al., 2019) | 15.6 / - | 27.0 / - | 35.9 / - | 39.7 / - | 37.9 / - |
| AET-project (Zhang et al., 2019) | 19.2 / 22.1 | 32.8 / 32.9 | 40.6 / 37.1 | 39.7 / 36.2 | 37.7 / 34.7 |
| DAE | 12.5 / 15.9 | 18.5 / 22.6 | 21.8 / 24.2 | 20.4 / 22.1 | 14.8 / 18.1 |
| DAE + Trans | 17.6 / 21.6 | 31.8 / 31.7 | 39.2 / 35.7 | 37.4 / 34.1 | 34.5 / 32.8 |
| (Ours) LapDAE | 18.4 / 21.0 | 27.4 / 30.9 | 29.9 / 31.6 | 27.0 / 29.2 | 22.7 / 26.1 |
| (Ours) LapDAE + Trans | **19.3 / 22.2** | **33.2 / 33.8** | **43.2 / 38.2** | **41.1 / 37.3** | **39.6 / 36.1** |

**Conv1 learned filter visualization:** In Fig. 5, we show the comparison for the learned filters from the first layer (*i.e.*, *conv1*) of AlexNet between our approach and the fully-supervised ones. In the supervised version (the left panel), both color blobs and edge filters are learned. We can see that although not as sharp as those filters learned by the supervised setup for some blobs, our approach (the right panel) learns quite good filters including edges along different directions, edges with different frequencies, color contrast along different directions, *etc.*, similar to the supervised ones. Comparing with conventional DAE (the middle panel), the learned representations are much better.

**Controlled classification:** Here we quantitatively evaluate our learned representations on the ImageNet classification task (Deng et al., 2009). Following the experimental settings in (Zhang et al., 2016), we freeze the pre-trained weights of our model and train a linear classifier on the top of each *conv* layer, to perform the 1000-category classification task. In order to have approximately the same dimensions across different layers, the feature maps of each layer is interpolated to have around 9000 elements. Table 1 shows the evaluation results. Several state-of-the-art self-supervised representation learning methods (Zhang et al., 2016; 2017; Pathak et al., 2016; Doersch et al., 2015; Noroozi & Favaro, 2016; Donahue et al., 2016; Noroozi et al., 2017; Gidaris et al., 2018; Ren & Lee, 2018; Wu et al., 2018; Zhang et al., 2019; Huang et al., 2019) are included for comparison. Since the proposed approach can be easily integrated into other representation learning frameworks, we also present the performance of our LapDAE combined with the task of predicting transformations (*LapDAE+Trans*). Specifically, we base on the AET framework (Zhang et al., 2019) while reasoning the transformation between the original image and a transformed one corrupted by our LapDAE.

Table 2: Comparison with state-of-the-art representation learning methods on PASCAL VOC vision tasks of classification, detection on 2007, and semantic segmentation on 2012. For classification we also compare a setup that fixes the layers before *conv5* and only train the *fc6-8*. The learned weights from unlabeled ImageNet are transferred for the new tasks except the *ImageNet labels*.

| Method | Classification (%mAP) | | Detection (%mAP) | Segmentation (%mIoU) |
|---|---|---|---|---|
| | fc6-8 | all | all | all |
| ImageNet labels | 78.9 | 79.9 | 56.8 | 48.0 |
| Random Gaussian | - | 53.3 | 43.4 | 19.8 |
| Random rescaled (Krähenbühl et al., 2015) | 39.2 | 56.6 | 45.6 | 32.6 |
| Context (Doersch et al., 2015) | - | 55.3 | 45.7 | - |
| Context Encoder (Pathak et al., 2016) | 34.6 | 56.5 | 44.5 | 29.7 |
| Colorization (Zhang et al., 2016) | 61.5 | 65.5 | 46.9 | 35.6 |
| Counting (Noroozi et al., 2017) | - | 67.7 | 51.4 | 36.6 |
| GAN (Goodfellow et al., 2014) | 40.5 | 56.4 | - | - |
| BiGAN (Donahue et al., 2016) | 52.3 | 60.1 | 46.9 | 34.9 |
| RotNet (Gidaris et al., 2018) | 70.9 | 73.0 | 54.4 | 39.1 |
| AET-project (Zhang et al., 2019) | 70.5 | 73.1 | 54.2 | 39.3 |
| DAE | 37.0 | 54.6 | 43.4 | 29.1 |
| DAE + Trans | 66.7 | 70.1 | 51.0 | 36.8 |
| (Ours) LapDAE | 50.6 | 59.0 | 45.6 | 38.3 |
| (Ours) LapDAE + Trans | **71.4** | **74.2** | **55.2** | **41.1** |

From the results, we observe that the proposed method with the Laplacian pyramid largely improves the performance compared to its counterpart without the Laplacian pyramid, especially for the lower convolutional layers. This is consistent with the above visualization and analysis of the *conv1* layer, where the filter kernels have more representative power in the proposed LapDAE. When incorporating with the transformation prediction task, we can see that the performance is further boosted by a large margin. Even when compared to the *AET-project* method, our approach performs much better, which again validates the effectiveness of the proposed LapDAE.

## 4.5 TRANSFER LEARNING ON PASCAL VOC AND PLACES

Next we perform a transfer learning evaluation on the PASCAL VOC dataset (Everingham et al., 2010), for the tasks of classification, detection on VOC 2007, and semantic segmentation on VOC 2012. The learned weights of our model trained on ImageNet are transferred to a standard AlexNet for the evaluation. We then fine-tune the model on the PASCAL VOC trainval set and test on the test set. Note that we do not apply any "magic" techniques such as weights rescaling (Krähenbühl et al., 2015). For the classification task, we use the same network architecture as in the ImageNet evaluation, while for the semantic segmentation tasks we use the publicly available frameworks FCN (Long et al., 2015) and follow the same setups for other state-of-the-art methods *e.g.*, (Doersch et al., 2015; Zhang et al., 2016). The results in Table 2 show that the learned visual features by the proposed LapDAE exhibit good performances when transferred to other vision datasets or tasks, and performs favorably against the other state-of-the-art methods. For the classification task of PASCAL VOC, the proposed LapDAE outperforms all other generic unsupervised learning methods like DAE and GAN. More impressively, its segmentation performance on PASCAL VOC surpasses most of the representation learning methods including those self-supervised learning approaches with specifically designed pretext tasks. Comparison between the proposed framework and its counterpart DAE suggests that the improvement is partly due to the Laplacian pyramid. When incorporating the transformation prediction task (*LapDAE+Trans*), our approach on PASCAL VOC transfer learning is further improved and achieves state-of-the-art performance.

In addition to PASCAL VOC, we further evaluate the representation learning capability by transfer learning on the Places dataset (Zhou et al., 2014). We use the same experimental settings as the ImageNet experiment and the result is shown in Table 1. Differing from the ImageNet experiments, the classifier trained on top of different layers is a 205-way logistic regression layer and which is then trained with the Places labels. From the result we can see that, the proposed LapDAE performs

better than its counterparts and outperforms other methods when incorporating with transformation prediction.

## 5 CONCLUSION

In this paper we introduced a novel type of denoising autoencoder for unsupervised representation learning. In contrast to conventional DAE, the corrupted data input to the proposed DAE is produced with the aid of a Laplacian pyramid. By adding corruptions to randomly chosen levels in a Laplacian pyramid, the resulting data corruptions spans multiple scales across the original data space. From this, the network is forced to learn to represent underlying data structures across multiple scales. The proposed learning framework ensures that the agent learns better representations when compared to a conventional DAE and other self-supervised learning methods. While in this paper we showcase the effectiveness of the proposed method for learning transferable representations for vision tasks, it would be interesting to see how it performs with other types of data. It is also interesting to explore the possibility of extending the proposed idea to other representations like wavelet, SIFT, and other deep features. The core idea of performing both local and non-local learning that is consistent with the hierarchical nature of ConvNets, is the Laplacian pyramid space, which we believe is a promising direction for future research.

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

## APPENDIX

Here we present more qualitative results to show the performance of the proposed method, in addition to those shown in Fig. 3 and Fig. 4 in the main paper. Specifically, we first show the performance on MNIST dataset in Fig. 6. Additional qualitative reconstruction performance on the CIFAR-10 dataset is presented in Fig. 7. We also show more results on the image retrieval task that is presented in Fig. 4 (b). The additional results on image retrieval is shown in Fig. 8. The additional qualitative results again validate the effectiveness of the proposed approach on self-supervised representation learning.

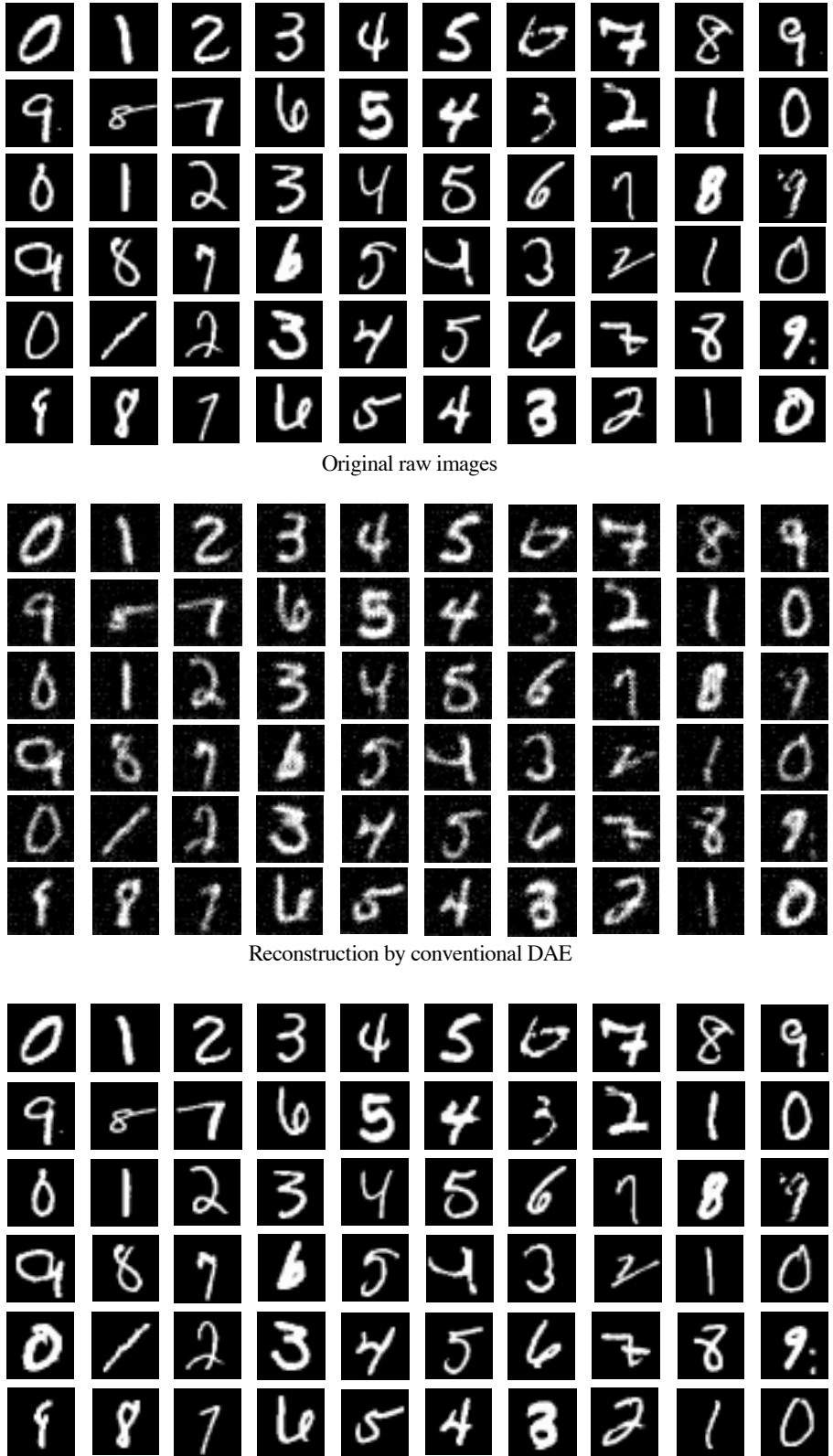

Figure 6: Additional results on illustration of the reconstruction performance using the MNIST dataset. The original raw input images are randomly selected from the test set and are shown in the first part, while the second and third parts show the reconstructed results from the conventional DAE and the proposed LapDAE, respectively.

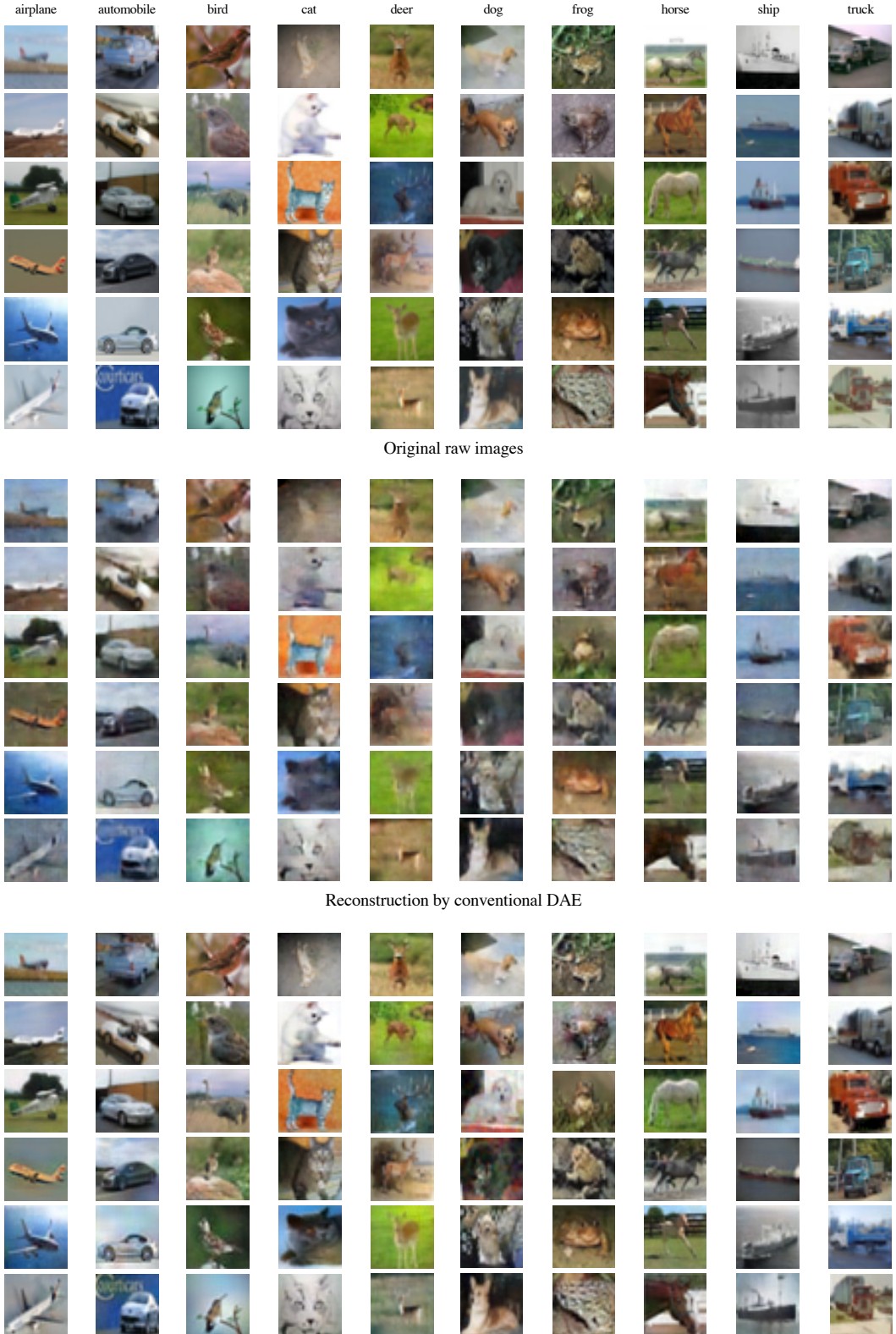

Figure 7: Additional reconstruction results comparison using the CIFAR-10 dataset, in which examples from each category in the test set are randomly selected for visualization. The original raw input images are shown in the first part, while the second and third parts show the reconstructed results from the conventional DAE and the proposed LapDAE, respectively.

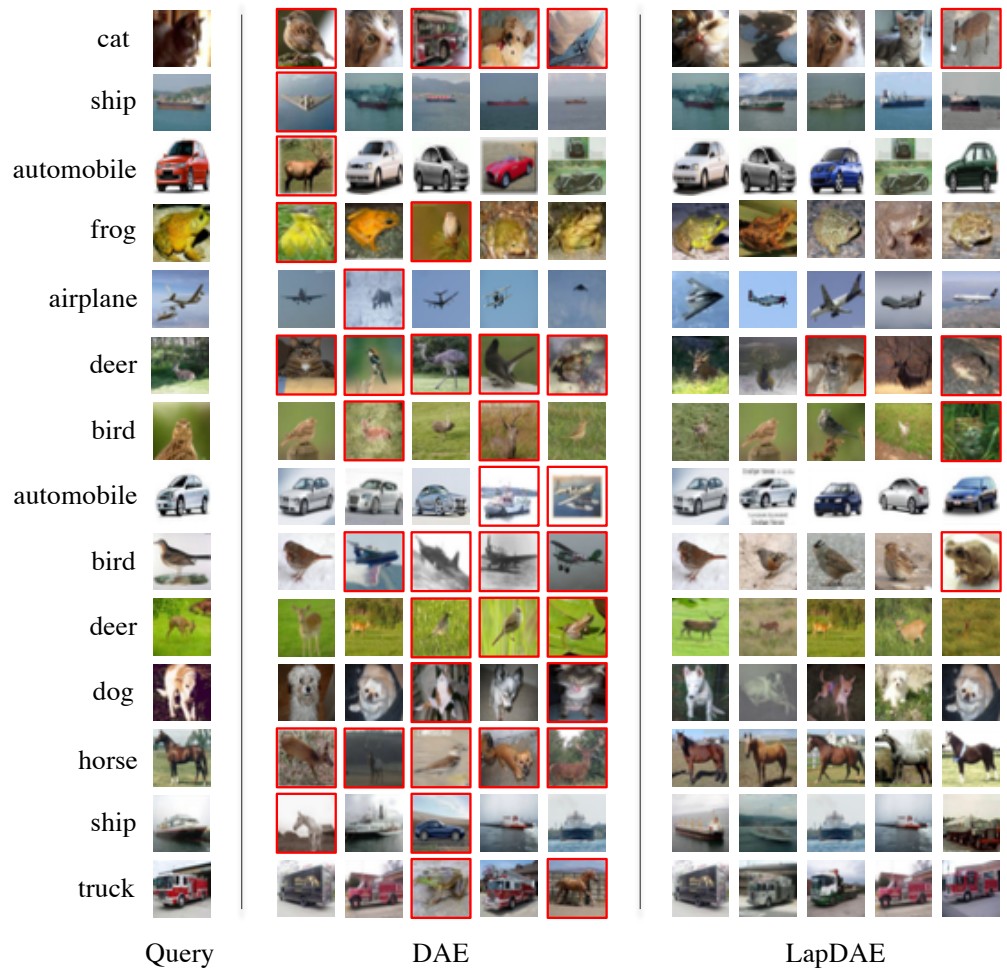

Figure 8: Additional image retrieval results using the CIFAR-10 dataset by nearest neighbor. Given the query on the left, the top-5 (from left to right) retrieved results of DAE (middle) and LapDAE (right) are presented, in which red marked ones indicate wrong category.

