# OpenReview forum: "Laplacian Denoising Autoencoder"
_ICLR.cc/2020/Conference — Reject_

### Official Review · AnonReviewer2 · 2019-10-15
**Official Blind Review #2**

**Rating:** 6

**Review:**

Denoising auto-encoder (DAE)  is a representation learning framework proposes in [1], which uses noisy input to reconstruct the clean “repaired” input. In the conventional DAE, the noise is directly added to the input space. This paper introduces a novel type of noise, which corrupts the Laplacian pyramid representation, and uses it to train a DAE. The effect of the proposed perturbation is to allow larger scale and semantically meaningful corruption, which may help to learn more transferable representation. Several experiments are conducted showing the effectiveness of the method
1) On MNIST, LapDAE provides better reconstruction images
2) On Cifar-10, LapDAE provides better image retrieval results
3) On Imagenet, combining with the transformation technique in [2], LapDAE achieve state-of-the-art result
4) On Pascal VOC,  combining with the transformation technique, LapDAE achieve state-of-the-art result on transfer learning
Overall, I find the idea natural and simple (simplicity of an approach is a good quality to me). At the same time, I also find the contribution a bit limited. The following are further comments and questions.

* From the experiments, standard DAE are harder to train comparing to the proposed LapDAE. In my opinion, this suggests that the local noise are harder to remove comparing to the Laplacian noise. It would be interesting to perform the following experiment to further understand the difference between DAE and LapDAE: train a DAE and LapDAE then
a) Apply LapDAE to reconstruct an image where the random noise is added on the input space (as the standard DAE setting).
b) Apply DAE to reconstruct an image where the noise is added to the Laplacian pyramid representation.

* How does the corrupted set C been selected? Moreover, how does the selected layer in the Laplacian pyramid effects the performance? For example, is there a difference in terms of performance of the model when training on LPS4 versus LPS8?

* The transformation technique seems to be very helpful for the performance, is it possible to combine it with other benchmark methods like RotNet or Counting?

[1]  Vincent et al. 2010, Stacked denoising autoencoders: Learning useful representations in a deep network with a local denoising criterion
[2] Zhang et al. 2019, AET vs. AED: Unsupervised Representation Learning by Auto-Encoding Transformations rather than Data

**Experience Assessment:**

I have read many papers in this area.

**Review Assessment: Checking Correctness Of Derivations And Theory:**

N/A

**Review Assessment: Checking Correctness Of Experiments:**

I assessed the sensibility of the experiments.

**Review Assessment: Thoroughness In Paper Reading:**

I read the paper thoroughly.

---

> ### Author Response · Authors · 2019-11-14
> **Response to Reviewer #2**
>
> Thank you for your positive comments and helpful suggestions. Our response to these suggestions are as follows:
>
> 1. From the experiments, standard DAE are harder to train comparing to the proposed LapDAE. In my opinion, this suggests that the local noise are harder to remove comparing to the Laplacian noise. It would be interesting to perform the following experiment to further understand the difference between DAE and LapDAE: train a DAE and LapDAE then
> a) Apply LapDAE to reconstruct an image where the random noise is added on the input space (as the standard DAE setting).
> b) Apply DAE to reconstruct an image where the noise is added to the Laplacian pyramid representation.
>
> Thanks for the suggestions. We provide the results of the two suggested settings in Fig. 3 (left) and Sec. 4.2 in the revision. Generally, when applying LapDAE to images with corruption on the input space (the first suggested setting), the reconstruction is slightly worse than LapDAE with corruption on the Laplacian space, while still much better than the conventional DAE. On the other hand, when applying DAE to images with corruption on the Laplacian space (the second suggested setting), the reconstruction is improved a bit but worse than the proposed LapDAE. Note that our LapDAE is not trained to denoise spatial noise on the input space but to capture more non-local context information. This is supported by the results where the digits are well reconstructed even with some minor noise in the background.
>
> 2. How does the corrupted set C been selected? Moreover, how does the selected layer in the Laplacian pyramid effects the performance? For example, is there a difference in terms of performance of the model when training on LPS4 versus LPS8?
>
> The corruption set C can be any types of corruptions theoretically. For simplicity, in this paper we choose the random Gaussian noise (Sec. 4.1). As shown in Fig. 1 (a) and discussed in Sec. 3.2, corruption applied to higher levels of the pyramid affects lager spatial scales of the image while that applied to lower levels affects more local area. In the proposed method, we randomly select the levels where the corruption applied to. As a result, the proposed model is able to capture both local and non-local representations through the network optimization. We empirically found that the model performs worse if only apply the corruption on a single level (e.g., LPS4 or LPS8) and it will degrade to the DAE if the corruption is applied at the lowest level (i.e., original image). The model trained on LPS4 performs slightly worse than that trained on LPS8 for deeper layer (e.g., conv5) representation learning while comparable for shallower layer (e.g., conv1).
>
> 3. The transformation technique seems to be very helpful for the performance, is it possible to combine it with other benchmark methods like RotNet or Counting?
>
> Yes, we believe our technique can be easily combined with other methods like RotNet and Counting, as the proposed technique makes no assumption of the pretext task and thus would not affect the pipeline of other benchmark methods. However, it is difficult to enumerate all possible combinations (each setting has to train a separate model for each dataset) in one single paper and is out-of-scope for our work. Due to the time limit, we perform an initial experiment of combining our technique with the RotNet. Till the time of the revision submission, the preliminary result shows that the proposed LapDAE is able to improve the performance of RotNet as well: (19.0, 31.9, 39.1, 38.6, 37.0) compared to (18.8, 31.7, 38.7, 38.2, 36.5) for the ImageNet performance in Table 1. We will update the final results once we finish the training.

---

> > ### Comment · AnonReviewer2 · 2019-11-14
> > **Re: Response to Reviewer #2**
> >
> > Thank you for the response and clarification!
> > I find the comparison newly added in Figure 3 very interesting, it suggests that the proposed LapDAE is more powerful compared to standard DAE because it could also handle local noise pretty well.  I thank you for taking time to add more experiments combining the transformation technique with RotNet. I believe this provides a stronger baseline and a fair comparison.
> > Overall, I like the idea and my concerns are carefully addressed. I will keep my score and discuss with other reviewers in favor of the acceptance.

---

### Official Review · AnonReviewer3 · 2019-10-22
**Official Blind Review #3**

**Rating:** 3

**Review:**

This paper provides a novel approach to learning useful representations with deep learning models. They formulate an entirely unsupervised framework based off autoencoders to accomplish this task. Their data differs from previous works by upsampling a noise-corrupted downsampling of the original inputs. The autoencoder is then trained to reconstruct the original image from this new data. Their experiments demonstrate that using this Laplacian pyramid scheme to generate noisy data leads to an autoencoder that learns better representations compared to a standard DAE.
Overall, the work in this paper has the potential to be a contribution to ICLR but lacks experimental completeness and clarity. Moreover, the main contribution is a better denoising autoencoder, but in the grand scheme of representation learning, it is unclear how broad of a contribution this is. I would be willing to change my score, upon addressing the following details:
•	As mentioned above, the results as presented provide a better DAE. The paper would be much stronger if it also provided comparisons to more recent models in representation learning closer to state of the art. For instance, they choose BiGAN as a model for comparison, but these were developed over three years ago and are now outdated in favor of better GAN models.
•	The paper would be further strengthened with additional experiments of their representations being qualitatively better than previous models. Their example of image retrieval via nearest neighbor is quite limited when compared to the wealth of tasks GAN models can accomplish.
•	In general, when presenting qualitative results (Figures 3,4, and 5), additional examples should be put into supplementary materials so as to demonstrate the paper did not cherry pick.
•	The presentation of the quantitative results is peculiar. The paper chooses to combine their Laplacian DAE with the AET framework. As such, all the results tables should include numbers for AET alone and the conventional DAE with AET.
•	Lastly, the paper needs a revision for ease of readability, as there are a significant number of grammatical errors that make it hard to read at times.
While the transfer learning experiments seem incomplete to me, that is not my area of expertise and I cannot judge how convincing that setup is as well as other reviewers.


**Experience Assessment:**

I have published one or two papers in this area.

**Review Assessment: Checking Correctness Of Derivations And Theory:**

I carefully checked the derivations and theory.

**Review Assessment: Checking Correctness Of Experiments:**

I assessed the sensibility of the experiments.

**Review Assessment: Thoroughness In Paper Reading:**

I read the paper thoroughly.

---

> ### Author Response · Authors · 2019-11-14
> **Response to Reviewer #3**
>
> Thank you for your positive comments and insightful suggestions. Our response to the comments and suggestions are provided below:
>
> 1. As mentioned above, the results as presented provide a better DAE. The paper would be much stronger if it also provided comparisons to more recent models in representation learning closer to state of the art. For instance, they choose BiGAN as a model for comparison, but these were developed over three years ago and are now outdated in favor of better GAN models.
>
> Thanks for the suggestion. As shown in our original submission in Table 1 and Table 2, we compared to several recent models such as RotNet [Gidaris et al. 2018], Domain Adapt [Ren & Lee 2018], and AET-project [Zhang et al. 2019]. As suggested, we further include more recent state-of-the-art works in our revision: Instance [Wu et al. 2018], AND [Huang et al. 2019].
>
> 2. The paper would be further strengthened with additional experiments of their representations being qualitatively better than previous models. Their example of image retrieval via nearest neighbor is quite limited when compared to the wealth of tasks GAN models can accomplish.
>
> Thanks for the suggestions. We took the image retrieval as an illustration of the effectiveness of the representation learning ability. Since our work is not a GAN-based solution and does not focus on image synthesis, we did not present those image synthesis tasks as in conventional GAN models. We believe our quantitative experiments including those on large-scale datasets demonstrate the effectiveness of the proposed approach in representation learning. Please kindly let us know if there are any other necessary qualitative experiments need to be added and we would be happy to provide.
>
> 3. In general, when presenting qualitative results (Figures 3,4, and 5), additional examples should be put into supplementary materials so as to demonstrate the paper did not cherry pick.
>
> As suggested, we provide additional examples for the qualitative results into the appendix of the revision. Please refer to Fig. 6 – Fig. 8 in the Appendix for more details.
>
> 4. The presentation of the quantitative results is peculiar. The paper chooses to combine their Laplacian DAE with the AET framework. As such, all the results tables should include numbers for AET alone and the conventional DAE with AET.
>
> In Table 1 of our original submission, we presented the performance for AET alone (the fourth line from the bottom). Since the authors did not present their performance on Pascal VOC in their paper, we re-implement it and provide the corresponding results in Table 2 in the revision. As suggested, we also add the results for the conventional DAE with AET in all the tables in the revision.
>
> 5. Lastly, the paper needs a revision for ease of readability, as there are a significant number of grammatical errors that make it hard to read at times.
>
> Thanks for pointing out this and sorry for the readability. We have carefully revised the writing and improved the readability in the revision.
>
>
> We performed the transfer learning experiments following prior works, e.g., [Doersch et al. 2015] [Zhang et al. 2016] [Noroozi et al. 2017] [Gidaris et al. 2018]. As a result, we believe the transfer learning experiments is complete to our knowledge.
>
> Please kindly let us know if we address your concerns.

---

### Official Review · AnonReviewer1 · 2019-10-24
**Official Blind Review #1**

**Rating:** 6

**Review:**

This paper proposes a denoising auto-encoder where the input image is corrupted by adding noises to its Laplacian pyramid representation. Then a DAE is trained to predict the original data and learn a good representation of the data. By corrupting the Laplacian representation, which is multi-scale, the corruption of the image is not local and thus more robust representations are learned.

I personally like this idea. However, it seems a simple extension of the classical DAE.

1. Is it possible to generalize this idea to other representations of the images such as wavelets or sift, or the representations learned by other neural networks? It seems that you can add corruptions to any image representations as long as you can reconstruct the image from the representation. Here, you can reconstruct from  Laplacian pyramid representation.

2. As an empirical work, the experiments in this work is rather small-scale, using only CIFAR10 and MNIST. That seems far from sufficient.

**Experience Assessment:**

I have published one or two papers in this area.

**Review Assessment: Checking Correctness Of Derivations And Theory:**

I carefully checked the derivations and theory.

**Review Assessment: Checking Correctness Of Experiments:**

I carefully checked the experiments.

**Review Assessment: Thoroughness In Paper Reading:**

I read the paper thoroughly.

---

> ### Author Response · Authors · 2019-11-14
> **Response to Reviewer #1**
>
> Thank you for your appreciation of our idea and the constructive suggestions. Our response to these suggestions are provided below:
>
> 1. Is it possible to generalize this idea to other representations of the images such as wavelets or sift, or the representations learned by other neural networks? It seems that you can add corruptions to any image representations as long as you can reconstruct the image from the representation. Here, you can reconstruct from Laplacian pyramid representation.
>
> Thanks for the suggestions. While these are interesting directions, they are out-of-scope for this paper. We update in the revision in Section 5 to include them as possible directions to explore. To the best of our knowledge, this work makes the first attempt towards Laplacian space manipulation for self-supervised representation learning and is shown to be effective on both small-scale and large-scale representation learning. We believe our work has the potential to shed light on further progress in this direction.
>
> 2. As an empirical work, the experiments in this work is rather small-scale, using only CIFAR10 and MNIST. That seems far from sufficient.
>
> There might be a misunderstanding here. In the original submission, in addition to the CIFAR10 and MNIST, we did perform experiments and evaluations on large-scale datasets like ImageNet, Places and Pascal VOC, as shown in Sec. 4.4 and Sec. 4.5. We believe these evaluations demonstrated the effectiveness of the proposed method in self-supervised representation learning.

---

### Decision · Program_Chairs · 2019-12-19

**Decision:**

Reject

**Comment:**

The main idea proposed by the work is interesting. The reviewers had several concerns about applicability and the extent of the empirical work. The authors responded to all the comments, added more experiments, and as reviewer 2 noted, the method is interesting because of its ability to handle local noise. Despite the author's helpful responses, the ratings were not increased, and it is still hard to assess the exact extent of how the proposed approach improves over state of the art.   Because some concerns remained, and due to a large number of stronger papers, this paper was not accepted at this time.